# Study on Nitrile Oxide for Low-Temperature Curing of Liquid Polybutadiene

**DOI:** 10.3390/ma15093396

**Published:** 2022-05-09

**Authors:** Ping Li, Xiaochuan Wang

**Affiliations:** 1School of Mechanical and Electronic Engineering, Guangzhou University, Guangzhou 510006, China; leeping@gzhu.edu.cn; 2School of Biomedical Sciences and Engineering, Guangzhou International Campus, South China University of Technology, Guangzhou 511442, China; 3National Engineering Research Centre for Tissue Restoration and Reconstruction, South China University of Technology, Guangzhou 510006, China; 4Key Laboratory of Biomedical Engineering of Guangdong Province, and Innovation Centre for Tissue Restoration and Reconstruction, South China University of Technology, Guangzhou 510006, China; 5Key Laboratory of Biomedical Materials and Engineering of the Ministry of Education, South China University of Technology, Guangzhou 510006, China

**Keywords:** low-temperature curing, tetramethyl-terephthalobisnitrile oxide, cross-linked rubber material

## Abstract

As a significant component of composite solid propellants, the cross-link alkenyl polymers need to cure at high temperatures and the current isocyanate curing systems are highly humidity sensitive. This paper presented a low-temperature curing method for a cross-linked polymer (polybutadiene) with stable wettability by using cycloaddition of the nitrile oxide of tetramethyl-terephthalobisnitrile oxide (TTNO) and the C=C group of liquid polybutadiene (PB). The TTNO was synthesized in four steps from 1,2,4,5-tetramethylbenzene and evaluated as a low-temperature hardener for curing liquid PB. To characterize the reaction ability of TTNO at 25 °C, the cross-linked rubber materials of various contents (8%, 10%, 12%, 14%, 16%) of curing agent TTNO were prepared. The feasibility of the curing method can be proved by the disappearance of the absorption peak of the nitrile oxide group (2300 cm^−1^) by FT-IR analysis. Contact angle, TG-DTA and tensile-test experiments were conducted to characterize the wettability, thermo-stability and mechanical properties of the obtained cross-linked rubber materials, respectively. The results showed that the curing agent TTNO could cure PB at room temperature. With the growing content of the curing agent TTNO, the tensile strength of the obtained cross-linked rubber material increased by 260% and the contact angle increased from 75.29° to 89.44°. Moreover, the thermo-stability performances of the cross-linked rubber materials have proved to be very stable, even at a temperature of 300 °C, by TGA analysis.

## 1. Introduction

Since the 1940s, people have used polymer in composite solid propellants for better mechanical properties [1,2]. Nowadays, the widely used polymer material is hydroxyl-terminated polybutadiene (HTPB). For example, the cross-linked polyurethane, which is synthesized by the hydroxyl-terminated polymer reacting with diisocyanate [3,4], is served as the main component of solid propellants because of its outstanding mechanical properties. However, the synthetic material system of isocyanate and hydroxyl-terminated polymer enables humidity-sensitive and toxic problems and needs high-temperature curing, which calls for a new curing system [5,6]. Recently, an eco-friendly binder has been developed by using a 1,2,3-triazole network system [7,8]. The 1,2,3-triazole network system can effectively promote the curing of azide binders, such as glycidyl azide polymer (GAP), poly (3-azidomethyl-3-methyl oxetane) (PAMMO), poly (3,3-diazidomethyloxetane) (PBAMO) and so on [9,10,11]. Nonetheless, it still needs a relatively high temperature to cure. 

Recent studies, however, show that a new curing system by using a 1,3-dipolar cycloaddition reaction of nitrile oxides instead of a 1,2,3-triazole network system can make the curing temperature lower [12,13]. Nitrile oxides are organic compounds that contain –CNO which can directly bind to the carbon atom [14,15]. The structure of the –CNO functional group offers higher polarity than the azide group [16,17], leading to a more activated reaction with the nitrile oxide, compared with the reaction with the azide [18,19]. Therefore, the sterically hindered bifunctional nitrile oxides can be used as curing agents to crosslink alkenyl polymers [20,21] at a lower temperature.

Tetramethyl-terephthalobisnitrile oxide (TTNO) is a stable crystalline solid and can be used as a room temperature curing agent [22,23,24]. There are two –CNO groups in each tetramethyl-terephthalobisnitrile oxide (TTNO) molecule. Liquid polybutadiene (abbreviated as liquid PB), also known as the low-molecular-weight polybutadiene, is a low-molecular-weight polybutadiene and is a liquid rubber without functional groups (except double bond) [25,26]. It is often used as a toughener for epoxy resin adhesives. 

In this study, a cross-linked polymer (rubber material) based on the C=C bond and TTNO was investigated under room temperature curing conditions. The reactivities of unsaturated bonds of liquid PB with TTNO to form isoxazoline were studied using Fourier transform infrared spectroscopy (FT-IR). The mechanical properties, fracture surfaces, thermal properties and wettability of cross-linked rubber material based on the liquid PB were reported.

## 2. Materials and methods

### 2.1. Materials

Hydrobromic acid (33 wt.% in acetic acid) and 2-nitropropane were obtained from Energy Chemical. Polybutadiene (predominantly a 1,2-addition, average Mn ~3000 by VPO) was obtained from Sigma-Aldrich (St. Louis, MO, USA). Analytical pure reagents, such as sodium hydroxide, hydroxylamine hydrochloride, potassium hydroxide, paraformaldehyde, and sodium hypochlorite were obtained from the Chengdu Kelong Chemical Company, Ltd. (Chengdu, China). Other reagents, such as dichloromethane, glacial acetic acid, isopropane, etc., were obtained from the Xi’an Chemical Reagents Factory (Xi’an, China). 

### 2.2. Synthesis of TTNO

The TTNO was synthesized from 1,2,4,5-tetramethylbenzene with four steps to obtain a total yield of 72%. The following formula shows the synthetic scheme (see Figure 1a), a similar synthetic reaction has also been reported in our previous work [27].

**1,4-Bis(bromomethyl)-2,3,5,6-tetramethylbenzene.** To a mixture of 1,2,4,5-tetramethylbenzene (18.0 g, 0.133 mol), paraformaldehyde (8.0 g, 0.267 mol) and 66.7 mL of glacial acetic acid were added to 53.3 mL of a 33 wt% HBr/acetic acid solution rapidly. The mixture was kept for 8 h at 120 °C and then poured into 100 mL of water. The product was filtered off on a G3 glass frit and dried in vacuum. The yield was 38.8 g (91%) of 1,4-bis(bromomethyl)-2,3,5,6-tetramethylbenzene as a white powder. The NMR test results: ^1^H NMR(500 MHz, CDCl_3_), δ 4.58(s, 4H), 2.32(s, 12H); ^13^C NMR(125 MHz, CDCl_3_), δ 134.6, 134.1, 30.9, 15.9. *Anal.* Calcd for C_12_H_16_Br_2_: C, 45.03; H, 5.04. Found: C, 45.12; H, 4.96.

**2,3,5,6-Tetramethylterephthalaldehyde.** To a stirred solution of potassium hydroxide (15.9 g, 0.246 mol) and 2-nitropropane (25.1 g, 0.246 mol) in isopropanol (650 mL) under nitrogen, 1,4- bis(bromomethyl)-2,3,5,6-tetramethylbenzene (38.0 g, 0.102 mol) was added. After stirring under nitrogen for 4 days, the solvent was removed in vacuum. The residue was dispersed in water and then filtered to give the aldehyde, as a white solid of 22.0 g (98%). The NMR test results: ^1^H NMR(500 MHz, CDCl_3_), δ 10.62(s, 2H), 2.35(s, 12H); ^13^C NMR(125 MHz, CDCl_3_), δ 196.8, 138.3, 134.7, 15.3. *Anal.* Calcd for C_12_H_14_O_2_: C, 75.76; H, 7.42. Found: C, 75.25; H, 7.31.

**Tetramethyl-terephthalaldioxime.** To a stirred solution of 2,3,5,6-tetramethylterephthalaldehyde (21.5 g, 0.113 mol) in alcohol (500 mL), hydroxylamine hydrochloride (31.97 g, 0.460 mol) and sodium hydroxide (18.80 g, 0.470 mol) was added at 50 °C. After stirring under reflux for 4 h, the solvent was removed in vacuum. The residue was dispersed in water and then filtered to give the oxime, as a white solid of 24.2 g (95%). The NMR test results are: ^1^H NMR(500 MHz, DMSO-d_6_), δ 11.15(s, 2H), 8.31(s, 2H), 2.18(s, 12H); ^13^C NMR(125 MHz, DMSO-d_6_), δ 149.1, 133.2, 132.5, 17.5. *Anal.* Calcd for C_12_H_16_N_2_O_2_: C, 65.43; H, 7.32; N, 12.72. Found: C, 65.63; H, 7.31; N, 11.98.

**Tetramethyl-terephthalobisnitrile oxide****(TTNO)****.** To a stirred mixture of tetramethyl-terephthalaldioxime (24.0 g, 0.109 mol) and dichloromethane (240 mL), 480 mL NaClO/H_2_O solution (10% available chlorine) was added dropwise at 0 °C. The mixture was stirred at room temperature for another 1 h. The organic layer was separated, and the solvent was removed in vacuum. The yield was 20.0 g (85%) of tetramethyl-terephthalobisnitrile oxide as a slightly yellowish powder. The NMR test results are: ^1^H NMR(500 MHz, CDCl_3_), δ 2.41(s, 12H); ^13^C NMR(125 MHz, CDCl_3_), δ 138.1, 117.5, 18.6. The result of the NMR is the same as the theoretical value. The value of elemental analysis: *Anal.* Calcd for C_12_H_12_N_2_O_2_: C, 66.65; H, 5.59; N, 12.96. Found: C, 67.21; H, 5.76; N, 11.90. The elemental analysis is also similar to the theoretical value. Thus, it can be concluded from the NMR test results that TTNO was indeed synthesized.

### 2.3. Materials Preparation of the Cross-Linked Rubber Material

Liquid PB was used to study the potential of TTNO as a curing agent in a low-temperature curing system. The curing agent, the prepolymer and the dichloromethane were mixed to prepare the rubber material. A curing agent was mixed with a prepolymer and dichloromethane. To get a good cross-linked system, the stoichiometric ratio was calculated. Then, the rubber material of various contents (8%, 10%, 12%, 14%, 16%) of curing agent TTNO was chosen as a preparation, respectively. Take 10% as an example, 12 g liquid PB was mixed with 1.2 g of TTNO and degassed for 30 min. The solution was then cast into a Teflon mold and cured at 25 °C for 3 days to get an elastomer. Although TTNO needs to be dissolved in dichloromethane before it reacts with PB, it is easy to handle and has less toxicity compared to the isocyanates. Figure 1b shows the formation of the cross-linked rubber material. It can be seen that the cross-linked rubber material is a three-dimensional network structure.

### 2.4. Characterizations

According to GB/T 528-2009, the mixture was poured into a dumbbell-shaped Teflon mold with a thickness of 4.0 mm for the tensile test. The test was carried out on a C45.504 computer-controlled electronic testing machine at a rate of 100 mm/min.

The structure of cross-linked polymers was analyzed by Fourier transform infrared spectroscopy (FT-IR). The FT-IR were recorded on a Nicolet 5700 FT-IR spectrophotometer by the KBr-pellet method and scanned from 4000 to 400 cm^−1^, with a resolution superior to 0.5 cm^−1^.

The scanning electron microscopy (SEM) images were obtained on a field emission scanning electron microscope (Merlin, Carl Zeiss AG, Oberkochen, Germany) after the sputter coating of gold on the specimen surface. The morphologies of the fracture surface images of cross-linked polymer were also obtained on this scanning electron microscope.

Static contact angle measurements were carried out on a DSA30 contact angle measuring instrument (Kruss Scientific Instruments Co., Ltd., Hamburg, Germany) by deposing a water drop on the surface of cross-linked polymer.

The thermal properties of cross-linked polymers were characterized by thermogravimetric analysis (TG, SDT Q160, TA Instruments, Milford, American) in a purified nitrogen atmosphere with a flow of 100 mL/min. 

## 3. Results and Discussion

### 3.1. FT-IR Spectra of the Cross-Linked Rubber Materials

The FT-IR spectra of the blend of PB and TTNO (10%), liquid PB and the obtained cross-linked rubber material are shown in Figure 2a. The FT-IR spectra are an efficient method to analyze the curing progress. The observed peak at 2800~3100 cm^−^^1^ was attributed to the stretching vibration of methylene C−H. The observed peak at 1654 cm^−^^1^ was attributed to the stretching vibration of C=C. The spectra showed that the absorption peak of the nitrile oxide group (2300 cm^−1^) disappeared, indicating the finish of the curing reaction [28].

From Figure 1, we have already known that there are many C=C bonds in the liquid polybutadiene molecule and only two nitrile oxide groups in each TTNO molecule. The C=C bonds of the liquid polybutadiene react with nitrile oxide groups of TTNO to get the isoxazoline. As a result, the FT-IR absorption peak of nitrile oxide groups disappears. The FT-IR spectra of various contents of curing agent TTNO of the cured polymers are shown in Figure 2b. The comparison of the spectra shows that an absorption peak of C=C (1654 cm^−1^) exists, which means there still contains some C=C bonds. As the content of curing agent TTNO increases, there is still a small absorption peak of the nitrile oxide group (2300 cm^−1^), indicating some TTNO material is not involved in the curing reaction. This can also be used to explain the reason why the tensile strength (16%) is decreased (see Figure 5a). 

### 3.2. Fracture Surfaces of the Cross-Linked Rubber Materials

The SEM images on the fracture surfaces of cross-linked polymers are analyzed to see the obtained material change of the curing reaction of the polybutadiene and the curing agent TTNO. The brittle fracture (quenching in liquid nitrogen) of the cured polymers with different contents of curing agent TTNO is shown in Figure 3. It is seen, that the brittle fracture surface becomes more and more regular with the increasing content of curing agent TTNO, except the content of curing agent TTNO is over 16%. The increase in the content of curing agent TTNO in the cross-linked polymer results in the reduction of ductile fracture and the increase of brittle fracture, and is referred to as the structure of the fracture surface. 

### 3.3. Contact Angles of the Cross-Linked Rubber Materials

The contact angles were measured to estimate the wettability of the rubber material. A larger contact angle means it is not easy for water to infiltrate into the rubber material [29]. Figure 4 shows that the contact angle increases from 75.29° to 89.44° with the content of curing agent TTNO increasing from 8% to 16%. It may attribute to the component of the rubber material—isoxazoline—which is hydrophobic to liquid polybutadiene backbones, formed with the increase in the content of the curing agent TTNO. The isoxazoline ring is a weak-polar group. Due to the hydrophobic groups on the surface of the cross-linked rubber material, the interaction of water with the surface is reduced accordingly. Thus, the cross-linked rubber material is more hydrophobic as the content of the curing agent TTNO increases. Moreover, the three-dimensional network structure of the cross-linked rubber material also contributes to the deceased water permeability of the surface. Therefore, it is hinted that the cross-linked rubber material is more stable with regard to the wettability and can reduce the humidity that is sensitive to the curing systems. 

### 3.4. Mechanical Properties of the Cross-Linked Rubber Materials

As the tensile strength is an important index of mechanical properties of polymers [30], the tensile tests were carried out to estimate the mechanical strength of the cross-linked rubber material. The tensile strength relationship of the cross-linked rubber materials with different TTNO contents is shown in Figure 5a. At first, with the increase of curing agent TTNO, the tensile strength of the rubber material increased incrementally. The maximum increment in the tensile strength of the rubber material is about 210% when the content of curing agent TTNO is 14%. This is because more cross-linking points will be generated in the cross-linked rubber material by increasing the curing agent content, resulting in a better tensile strength performance from the macroscopic scale. However, when the curing agent is increased to 16%, the tensile strength is then decreased. This is because there are two nitrile oxide groups in each TTNO molecule and with the increasing of content of TTNO, more isoxazolines in the rubber material are formed and some nitrile oxide groups were not taking part in the curing reaction. This indicates that the decrease in effective cross-linking point will reduce the tensile strength of the rubber material.

### 3.5. Thermal Properties of the Cross-Linked Rubber Materials

Figure 5b shows the TG curves of the rubber material with various TTNO content. The thermal decomposition temperature of the rubber material can be seen from the TGA analysis. The rubber material begins to decompose at about 250 °C. The decomposition of the rubber material starts speeding up significantly around the temperature of 300 °C. The greatest weight difference of 2.2% in decomposition occurs at 375 °C for the cross-linked rubber materials with the TTNO contents between 10% and 16%; see the enlarged view in Figure 5b. When the temperature reaches 500 °C, the process of decomposition of the rubber material ends. It is worth noting, however, that the rubber material cured by TTNO with a content of 16% is not as stable as the other contents. It may be attributed to the temperature-sensitive component of isoxazoline, which is the product of the uncompleted reaction of TTNO and C=C bonds of PB. Therefore, the thermal stability of the cross-linked rubber material is dependent on the content of the uncompleted reaction product of isoxazoline.

The temperatures of weight loss of 5% and weight loss of 10% of the rubber material of various TTNO content are given in Table 1. As illustrated in Table 1, the rubber material with a TTNO content of 8–14% has higher thermal stability than the TTNO content of 16%. Moreover, the temperature of weight loss decreases as the TTNO content increases. This means that the rubber material with lower TTNO content, such as 8–12%, performs better in thermal stability. The results are also in accordance with the conclusion shown in Figure 5b.

## 4. Conclusions

Tetramethyl-terephthalobisnitrile oxide (TTNO) was synthesized from 1,2,4,5-tetramethylbenzene with four steps, as a low-temperature curing agent to generate the cross-linked rubber material based on liquid polybutadiene. The formation of the cross-linked network and the finish of the curing reaction were confirmed by the disappearance of –CNO signals in FT-IR. When the content of curing agent TTNO is 14%, the cross-linked rubber material shows the highest tensile strength with an increment of about 210%. The contact angle experiments of the rubber materials showed that the contact angle increased with the increased content of curing agent TTNO, resulting in a more stable wettability and better humidity robustness to the curing systems. Moreover, the performance of the cross-linked rubber material’s thermo-stability is very stable, even at a temperature of 300 °C.

## Figures and Tables

**Figure 1 materials-15-03396-f001:**
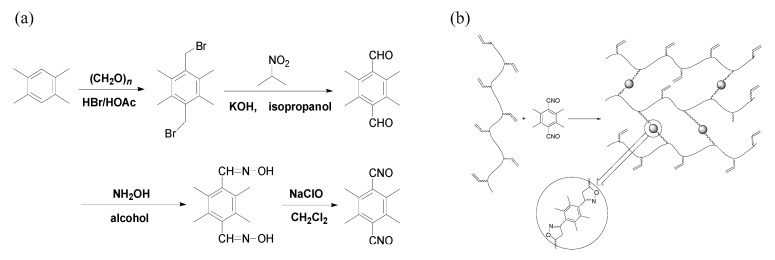
Synthesis of rubber material: (**a**) synthesis of TTNO and (**b**) formation of the cross-linked rubber material.

**Figure 2 materials-15-03396-f002:**
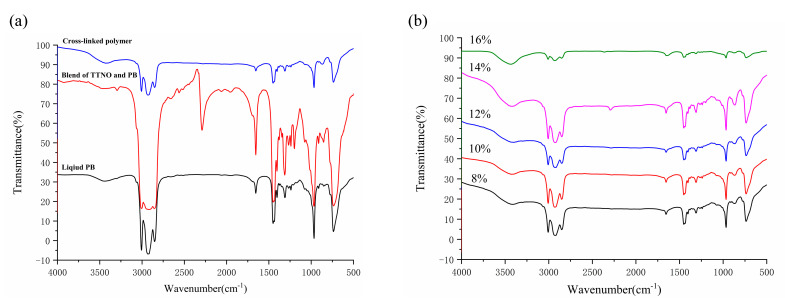
FT-IR spectra of (**a**) cross-linked polymer and the blend of PB and TTNO, (**b**) cross-linked polymers of different TTNO contents.

**Figure 3 materials-15-03396-f003:**
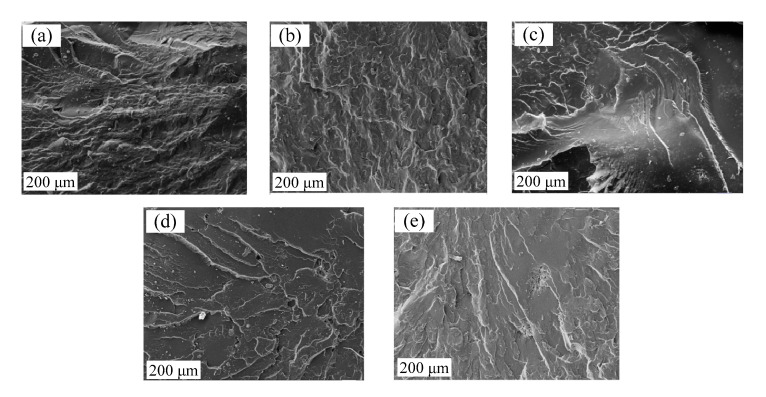
Brittle fracture surface of (**a**) 8% curing agent, (**b**) 10% curing agent, (**c**) 12% curing agent, (**d**) 14% curing agent, (**e**) 16% curing agent.

**Figure 4 materials-15-03396-f004:**
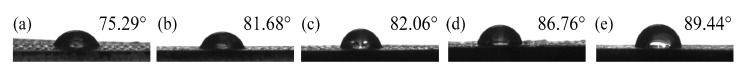
Contact angle of (**a**) curing agent 8%, (**b**) curing agent 10%, (**c**) curing agent 12%, (**d**) curing agent 14%, (**e**) curing agent 16%.

**Figure 5 materials-15-03396-f005:**
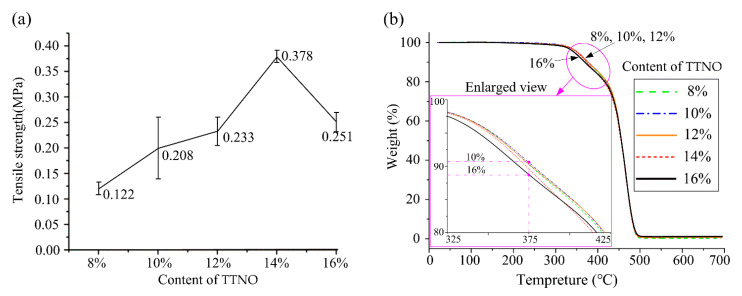
Properties of the rubber material of various TTNO content: (**a**) tensile strength and (**b**) TG curves.

**Table 1 materials-15-03396-t001:** TG data of the rubber material of various TTNO content.

TTNO Content	8%	10%	12%	14%	16%
T_5%_(°C)	352.23	352.74	351.93	350.52	344.11
T_10%_(°C)	376.17	377.65	376.57	373.27	368.98

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
