# Peer review of "Study on Nitrile Oxide for Low-Temperature Curing of Liquid Polybutadiene"

_materials, 2022, doi:10.3390/ma15093396_

Round 1

Reviewer 1 Report

The authors have synthesized Tetramethyl-terephthalobisnitrile oxide (TTNO) in order to serve as low-temperature hardener for polybutadiene at different percentages- 8,10,12,14,16. Their FTIR and TGA results indicate curing of the polybutadiene at room temperatures. Furthermore, mechanical results show minor improvement in tensile properties and an increase in contact angle. 

There are several issues pertaining to this manuscript: 

1)The entire manuscript needs a thorough revision in terms of grammar and spelling. 

2) The introduction is inadequate, not sufficient in terms of literature review, and does not clearly indicate the scientific merit of the author's approach.

3) The methods section does not give any details on the FTIR, thermal, mechanical, and contact angel tests. What standards were used for these tests ? What were the testing conditions and how many samples were tested ?

4) Figure 5 TGA graphs need to more clear, and symbols needs to be added. 

5) Further IR or NMR tests are highly suggested to confirm that TTNO was indeed synthesized. 

6) What were the cross-linking densities of these composites ?

7) The scientific merit and novelty of this work do not come across clearly to the audience. More detailed discussions need to be incorporated on where TTNO stands with respect to other approaches like 1,2,3-triazole network system and 1,3-dipolar cycloaddtion of nitrile oxides in this domain

Author Response

Reviewer 1

The authors have synthesized Tetramethyl-terephthalobisnitrile oxide (TTNO) in order to serve as low-temperature hardener for polybutadiene at different percentages- 8,10,12,14,16. Their FTIR and TGA results indicate curing of the polybutadiene at room temperatures. Furthermore, mechanical results show minor improvement in tensile properties and an increase in contact angle.

There are several issues pertaining to this manuscript:

1)The entire manuscript needs a thorough revision in terms of grammar and spelling.

Response:the manuscript has been thoroughly revised in terms of grammar and spelling.

2) The introduction is inadequate, not sufficient in terms of literature review, and does not clearly indicate the scientific merit of the author's approach.

Response: there are relatively few available literatures on nitrile oxide curing the polymers with double bonds so far. We have added relevant references published in recent years.

TTNO is stable crystalline solid and can be used as room temperature curing agent[22-24]. There are two –CNO groups in each tetramethyl-terephthalobisnitrile oxide (TTNO) molecule. Liquid polybutadiene, also known as low-molecular-weight polybutadiene, is a low-molecular-weight polybutadiene, which is a liquid rubber without functional groups(except double bond), abbreviated as liquid PB[25-26]. It is often used as a toughener for epoxy resin adhesives.

As a significant component of composite solid propellants, the crosslink alkenyl polymers need to cure at high temperature and the current isocyanate curing systems are highly humidity sensitive. The scientific merit and novelty of this work is presenting a low-temperature curing method for a cross-linked polymer (polybutadi-ene) with stable wettability by using cycloaddtion of the nitrile oxide of tetrame-thyl-terephthalobisnitrile oxide (TTNO) and the C=C group of liquid polybutadiene (PB).

3) The methods section does not give any details on the FTIR, thermal, mechanical, and contact angel tests. What standards were used for these tests ? What were the testing conditions and how many samples were tested ?

Response:The details on the methods have been added in the revised manuscript in Part 2.4 Characterizations.

According to GB/T 528-2009, the mixture was poured into a dumbbell-shaped Teflon mold with a thickness of 4.0 mm for tensile test. The test was carried out on a C45.504 computer-controlled electronic testing machine at a rate 100 mm/min. Three samples per group were tested.

The structure of cross-linked polymers were analyzed by the Fourier transform infrared spectroscopy(FT-IR). The FT-IR were recorded on a Nicolet 5700 FT-IR spectrophotometer by the KBr-pellet method and scanned from 4000 to 400 cm-1, with a resolution superior to 0.5 cm-1.

   The scanning electron microscopy(SEM) images were obtained on a field emission scanning electron microscope(Merlin, Zeiss company, Germany) after the sputter coating of gold on the specimen surface. The morphologies of the fracture surface im-ages of cross-linked polymer were also obtained on this scanning electron microscope.    Static contact angle measurements were carried out on a DSA30 contact angle measuring instrument (Kruss, Germany) by deposing a water drop on the surface of cross-linked polymer.

The thermal properties of cross-linked polymers were characterized by thermogravimetric analysis (TG, SDT Q160, TA company, American) in a purified nitrogen atmosphere with a flow of 100 mL/min.

4) Figure 5 TGA graphs need to more clear, and symbols needs to be added.

Response: Figure 5 has been revised and the symbols have been added.

5) Further IR or NMR tests are highly suggested to confirm that TTNO was indeed synthesized.

Response: The NMR and elemental analysis tests has been added. The NMR test results are: 1H NMR(500 MHz, CDCl3), δ 2.41(s, 12H); 13C NMR(125 MHz, CDCl3), δ 138.1, 117.5, 18.6. The result of NMR is the same with the theoretical value. And the value of elemental analysis: Anal. Calcd for C12H12N2O2: C, 66.65; H, 5.59; N, 12.96; the Found: C, 67.21; H, 5.76; N, 11.90. The elemental analysis is also similar with the theoretical value. Thus, it can be concluded from the NMR test results that TTNO was indeed synthesized. This section has been added in the revised manuscript in Part 2.2 Synthesis of TTNO.

6) What were the cross-linking densities of these composites ?

Response:In our previous studies, we have proved that the gel percentage of the similar elastomer was still greater than 90% After extraction with acetone or chloroform for 24 h.(Wang X C, Lu X M, Shu Y J, et al. 2019. Synthesis and curing of alkenyl polyether energetic binder. Fine Chem, 36, pp.348-353.)This means these composites may also have high cross-linking densities.

7) The scientific merit and novelty of this work do not come across clearly to the audience. More detailed discussions need to be incorporated on where TTNO stands with respect to other approaches like 1,2,3-triazole network system and 1,3-dipolar cycloaddtion of nitrile oxides in this domain

Response:As a significant component of composite solid propellants, the crosslink alkenyl polymers used to cure at high temperature and be humidity sensitive. This paper presented a method for curing a cross-linked polymer (rubber material) by using cycloaddtion of the nitrile oxide of TTNO and the C=C group of liquid PB and achieved low-temperature curing of liquid polybutadiene. Besides, the proved wetting stability of the obtained crosslink alkenyl polymer (polybutadiene) was also proved by contact angle test experiments.

We have added the detailed discussions about where TTNO stands with respect to other approaches like 1,2,3-triazole network system and 1,3-dipolar cycloaddtion of nitrile oxides in this domain in the revised manuscript in Part 1. Introduction.

TTNO is stable crystalline solid and can be used as room temperature curing agent[22-24]. There are two –CNO groups in each tetramethyl-terephthalobisnitrile oxide (TTNO) molecule. Liquid polybutadiene, also known as low-molecular-weight polybutadiene, is a low-molecular-weight polybutadiene, which is a liquid rubber without functional groups(except double bond), abbreviated as liquid PB[25-26]. It is often used as a toughener for epoxy resin adhesives.

[22] Wang X, Li P, Lu X, et al. 2021. Synthesis and curing of allyl urethane NIMMO-THF copolyether with three functional groups as a potential energetic binder. Cent Eur J Energ Mater, 2020, 17, pp142-163.

[23] Wang X, Li P, Lu X, et al. 2020. Synthesis and curing of AUT-PNIMMO with three functional groups. Polym-Korea, 43, pp.503-511.

[24] Fan Y, Tang C,  Hu Q, et al. 2018. Evaluation on curing properties and kinetics of isophthalonitrile oxide. Pol J Chem Technol, 20, pp.37-46.

[25]Vikram G, Johannes M, Muhammad T, et al. 2022. Development of liquid diene rubber based highly deformable interactive fiber-elastomer composites. Materials, 15, pp.390.

[26]Krzysztof P, Małgorzata M, Joanna P, et al. 2021. The effect of liquid rubber addition on the physicochemical properties, cytotoxicity, and ability to inhibit biofilm formation of dental composites. Materials, 14, pp.1704.

Reviewer 2 Report

The paper deals with the curing of polybutadiene.

It could have a sense and be published after the necessary modifications.

  • Abstract needs to highlight the main funding.
  • Introduction needs to be enhanced with 1-2 paragraphs stating the problems to be solved in the present paper and referencing scientific literature from 2020-to 2022 years.
  •  The sections 2.1 and 2.2 are copy-pasted. There is no synthesis description. Preparation details need to be given.
  • TTNO half lifetime is missing.
  • 3.1 section should have more details on FTIR groups identification. It is not clear, because Figure 2 was not discussed.
  • There is a lack of critical discussion in other sections considering Figures 3-5. A solid explanation of the data is missing.

It is recommended to enhance the discussions of data that is lacking in the manuscript.

Author Response

Reviewer 2

The paper deals with the curing of polybutadiene.

It could have a sense and be published after the necessary modifications.

  1. Abstract needs to highlight the main funding.

Response:The abstract has been rewritten with highlight the main finding as follows:

As a significant component of composite solid propellants, the crosslink alkenyl polymers need to cure at high temperature and the current isocyanate curing systems are highly humidity sensitive. This paper presented a low-temperature curing method for a cross-linked polymer (polybutadiene) with stable wettability by using cycloaddtion of the nitrile oxide of tetramethyl-terephthalobisnitrile oxide (TTNO) and the C=C group of liquid polybutadiene (PB). The TTNO was synthesized in four steps from 1,2,4,5-tetramethylbenzene and evaluate as a low-temperature hardener for curing the liquid PB. To characterize the reaction ability of TTNO at 25℃, the cross-linked rubber materials of various content (8%, 10%, 12%, 14%, 16%) of curing agent TTNO were prepared. The feasibility of the curing method can be proved by the disappearance of the absorption peak of nitrile oxide group (2300 cm-1) by FT-IR analysis. Contact angle, TG-DTA and tensile-test experiments were conducted to characterize the wettability, thermo-stability and mechanical property of the obtained cross-linked rubber materials, respectively. The results showed that the curing agent TTNO could cure PB under room temperature. With the growing content of curing agent TTNO, the tensile strength of the obtained cross-linked rubber material increased from about 0.122 MPa to 0.378 MPa and the contact angle increases from 75.29° to 89.44°. Besides, the thermo-stability performances of the cross-linked rubber materials have proved to be very stable even at the temperature of 300℃ by TGA analysis.

  1. Introduction needs to be enhanced with 1-2 paragraphs stating the problems to be solved in the present paper and referencing scientific literature from 2020-to 2022 years.

Response:As a significant component of composite solid propellants, the crosslink alkenyl polymers need to cure at high temperature and the current isocyanate curing systems are highly humidity sensitive. in terms of this issue, we carried out research and revised the introduction. The present paper and referencing scientific literature from 2020-to 2022 years has been added in the revised manuscript as refs [22]-[26].

[22] Wang X, Li P, Lu X, et al. 2021. Synthesis and curing of allyl urethane NIMMO-THF copolyether with three functional groups as a potential energetic binder. Cent Eur J Energ Mater, 2020, 17, pp142-163.

[23] Wang X, Li P, Lu X, et al. 2020. Synthesis and curing of AUT-PNIMMO with three functional groups. Polym-Korea, 43, pp.503-511.

[24] Fan Y, Tang C,  Hu Q, et al. 2018. Evaluation on curing properties and kinetics of isophthalonitrile oxide. Pol J Chem Technol, 20, pp.37-46.

[25]Vikram G, Johannes M, Muhammad T, et al. 2022. Development of liquid diene rubber based highly deformable interactive fiber-elastomer composites. Materials, 15, pp.390.

[26]Krzysztof P, Małgorzata M, Joanna P, et al. 2021. The effect of liquid rubber addition on the physicochemical properties, cytotoxicity, and ability to inhibit biofilm formation of dental composites. Materials, 14, pp.1704.

  1. The sections 2.1 and 2.2 are copy-pasted. There is no synthesis description. Preparation details need to be given.

Response:The sections 2.1 and 2.2 are revised. The details about preparation have been added in the revised manucscript in Part 2.2 Synthesis of TTNO.

1,4-Bis(bromomethyl)-2,3,5,6-tetramethylbenzene. To a mixture of 1,2,4,5- tetramethylbenzene (18.0 g, 0.133 mol), paraformaldehyde(8.0 g, 0.267mol), and 66.7 mL of glacial acetic acid was added 53.3 mL of a 33 wt% HBr/acetic acid solution rapidly. The mixture was kept for 8 h at 120℃and then poured into 100 mL of water. The product was flitered off on a G3 glass frit and dried in vacuum. The yield was 38.8 g(91%) of 1,4-bis(bromomethyl)-2,3,5,6-tetramethylbenzene as a white powder. The NMR test results: 1H NMR(500 MHz, CDCl3), δ 4.58(s, 4H), 2.32(s, 12H); 13C NMR(125 MHz, CDCl3), δ 134.6, 134.1, 30.9, 15.9. Anal. Calcd for C12H16Br2: C, 45.03; H, 5.04. Found: C, 45.12; H, 4.96.

2,3,5,6-Tetramethylterephthalaldehyde. To a stirred solution of potassium hydroxide(15.9 g, 0.246 mol), and 2-nitropropane(25.1 g, 0.246mol) in isopropanol(650 mL) under Nitrogen, 1,4- bis(bromomethyl)-2,3,5,6-tetramethylbenzene(38.0 g, 0.102mol) was added. After stirring under nitrogen for 4 days, the solvent was removed in vacuum. The residue was dispersed in water and then filtered to give the aldehyde, as a white solid 22.0 g(98%). The NMR test results: 1H NMR(500 MHz, CDCl3), δ 10.62(s, 2H), 2.35(s, 12H); 13C NMR(125 MHz, CDCl3), δ 196.8, 138.3, 134.7, 15.3. Anal. Calcd for C12H14O2: C, 75.76; H, 7.42. Found: C, 75.25; H, 7.31.

Tetramethyl-terephthalaldioxime. To a stirred solution of 2,3,5,6-tetramethylterephthalaldehyde (21.5 g, 0.113 mol) in alcohol (500 mL) was added hydroxylamine hydrochloride(31.97 g, 0.460 mol), and sodium hydroxide(18.80 g, 0.470 mol) at 50℃. After stirring under reflux for 4 h, the solvent was removed in vacuum. The residue was dispersed in water and then filtered to give the oxime, as a white solid 24.2 g(95%); The NMR test results: 1H NMR(500 MHz, DMSO-d6), δ 11.15(s, 2H), 8.31(s, 2H), 2.18(s, 12H); 13C NMR(125 MHz, DMSO-d6), δ 149.1, 133.2, 132.5, 17.5. Anal. Calcd for C12H16N2O2: C, 65.43; H, 7.32; N, 12.72. Found: C, 65.63; H, 7.31; N, 11.98.

Tetramethyl-terephthalobisnitrile oxide (TTNO).. To a stirred mixture of tetramethyl-terephthalaldioxime (24.0 g, 0.109 mol) and dichloromethane (240 mL) was added dropwise 480 mL NaClO/H2O solution(10% available chlorine) at 0℃. The mixture was stirred at room temperature for another 1 h. The organic layer was separated and the solvent was removed in vacuum. The yield was 20.0 g(85%) of tetramethyl-terephthalobisnitrile oxide as a slightly yellowish power. The NMR test results are: 1H NMR(500 MHz, CDCl3), δ 2.41(s, 12H); 13C NMR(125 MHz, CDCl3), δ 138.1, 117.5, 18.6. The result of NMR is the same with the theoretical value. And the value of elemental analysis: Anal. Calcd for C12H12N2O2: C, 66.65; H, 5.59; N, 12.96. Found: C, 67.21; H, 5.76; N, 11.90. The elemental analysis is also similar with the theoretical value. Thus, it can be con-cluded from the NMR test results that TTNO was indeed synthesized.

  1. TTNO half lifetime is missing.

Response:TTNO is a stable crystalline solid and we do not test the half lifetime.

  1. 1 section should have more details on FTIR groups identification. It is not clear, because Figure 2 was not discussed.

Response:the details on FTIR groups identification have been added as “The FT-IR spectra is an efficient method to analyse the curing progress. The observed peak at 2800~3100 cm–1 was attributed to the stretching vibration of methylene C–H. The observed peak at 1654 cm–1 was attributed to the stretching vibration of C=C. ”

Figure 2 has been discussed in the revised manuscript see Part 3.1.

  1. There is a lack of critical discussion in other sections considering Figures 3-5. A solid explanation of the data is missing.

Response:The discussions about Figures 3-5 have been added in the revised manuscript see Part 3.2-Part 3.5.

  1. It is recommended to enhance the discussions of data that is lacking in the manuscript.

Response:the discussions of data have been added in the revised manuscript.

Round 2

Reviewer 1 Report

All the proposed changes are acceptable. 

Figure 5 still needs to be revised with symbols. 

Author Response

All the proposed changes are acceptable.

Response: Thanks for the reviewer’s careful and professional review.

Figure 5 still needs to be revised with symbols.

Response: Figure 5 has been modified as the reviewer’s suggestion in the revised manuscript, and some discussions referring to figure 5b have also been added, see the contents marked in blue in section 3.5.

Reviewer 2 Report

Could be accepted

Author Response

Could be accepted.

Response: Thanks for the reviewer’s careful and professional review.